# Chronotype-Dependent Sleep Loss Is Associated with a Lower Amplitude in Circadian Rhythm and a Higher Fragmentation of REM Sleep in Young Healthy Adults

**DOI:** 10.3390/brainsci13101482

**Published:** 2023-10-19

**Authors:** Charlotte von Gall, Leon Holub, Martina Pfeffer, Simon Eickhoff

**Affiliations:** 1Institute of Anatomy II, Medical Faculty, Heinrich Heine University Düsseldorf, 40225 Düsseldorf, Germany; mail@leonholub.com (L.H.); martina.pfeffer@uni-duesseldorf.de (M.P.); 2Institute of Systems Neuroscience, Medical Faculty, Heinrich Heine University Düsseldorf, 40225 Düsseldorf, Germany; simon.eickhoff@uni-duesseldorf.de; 3Institute of Neuroscience and Medicine, Brain & Behaviour (INM-7), Research Centre Jülich, 52425 Jülich, Germany

**Keywords:** sleep architecture, chronotype, circadian, wearables, digital markers, sleep disturbances, sleep quality, Fitbit

## Abstract

In modern society, the time and duration of sleep on workdays are primarily determined by external factors, e.g., the alarm clock. This can lead to a misalignment of the intrinsically determined sleep timing, which is dependent on the individual chronotype, resulting in reduced sleep quality. Although this is highly relevant given the high incidence of sleep disorders, little is known about the effect of this misalignment on sleep architecture. Using Fitbit trackers and questionnaire surveys, our study aims to elucidate sleep timing, sleep architecture, and subjective sleep quality in young healthy adults (*n* = 59) under real-life conditions (average of 82.4 ± 9.7 days). Correlations between variables were calculated to identify the direction of relationships. On workdays, the midpoint of sleep was earlier, the sleep duration was shorter, and tiredness upon waking was higher than on free days. A higher discrepancy between sleep duration on workdays and free days was associated with a lower stability of the circadian rhythm of REM sleep and also with a higher fragmentation of REM sleep. Similarly, a higher tiredness upon waking on free days, thus under intrinsically determined sleep timing conditions, was associated with a lower proportion and a higher fragmentation of REM sleep. This suggests that the misalignment between extrinsically and intrinsically determined sleep timing affects the architecture of sleep stages, particularly REM sleep, which is closely connected to sleep quality.

## 1. Introduction

Sleep plays a crucial role in health in general and in mental health in particular. Insufficient sleep affects the immune system [1] and has far-reaching detrimental effects on higher brain function such as attention-, alertness- and emotion-related cognitive performance [2]. Sleep problems, which are highly prevalent in modern society, are associated with multiple adverse health consequences, including an increased risk of Alzheimer’s disease [3]. Importantly, individuals differ in their preferred midpoint of sleep (MS), defining the intrinsically determined chronotype [4,5] which is closely related to preference for morning or evening activity. Moreover, polymorphism in the circadian clock gene *PERIOD3*, which is associated with diurnal preference and delayed sleep phase syndrome [6], predicts individual differences in sleep architecture and susceptibility to decrement in cognitive performance induced by acute sleep loss [7]. In modern society, sleep duration (SD) and MS on workdays are mainly defined by the alarm clock in the morning; this not only ends sleep prematurely but also shifts the MS forward. Thus, early educational/work schedules result in a discrepancy in MS and SD between workdays and free days, which are defined as social jetlag and sleep loss, respectively [8]. The misalignment between socially determined sleep timing (on workdays) and intrinsically/biologically determined sleep timing (on free days) and has various health consequences, including impaired sleep quality and depressive symptoms [9,10]. The lower sleep quality on workdays than on free days is likely due to the interference of early schedules with the chronotype [11]. However, how this affects sleep architecture and circadian rhythms of sleep stages is still poorly understood.

Polysomnography, which is usually performed under laboratory conditions, is the gold standard for analysing sleep architecture. Young healthy subjects show 4–6 sleep cycles per night of alternate sleep stages including rapid eye movement (REM) sleep, light sleep (non-REM sleep stages 1 and 2) and deep sleep (non-REM sleep stages 3 and 4), interrupted by short bouts of waking. Deep sleep occurs mainly during the first half of sleep, while REM sleep predominates in the second half [12,13]. Disturbances in regular sleeping patterns are associated with pathological conditions such as sleep paralysis [14] and REM sleep behaviour disorder [15,16]. Fitbit wearables contain accelerometers and photoplethysmography sensors and Fitbit applies proprietary algorithms to detect sleep parameters and sleep stages based on movement and heart-rate patterns [17]. Fitbit wearables are inferior to polysomnography in terms of accuracy. Fitbit Alta, the predecessor of Fitbit Inspire used in this study, shows an overestimation of total sleep time (11.6 min), sleep efficiency (1.98%) and duration of deep sleep (18.2 min), while sensitivity/specificity are 0.73/0.72 for light sleep, 0.67/0.92 for deep sleep and 0.74/0.93 for REM sleep [18]. However, Fitbit wearables have the great advantage of enabling longitudinal analysis of human sleep recordings under real-life conditions, which is imperative for elucidating differences in sleep timing and architecture between workdays and free days. Moreover, digital phenotyping and optimization of the wearable algorithms for sleep quality and sleep disturbances offers great opportunities for broad use in clinical settings [19]. A recent study used real-life Fitbit sleep data in combination with questionnaire survey in a small and relatively homogenous sample of young healthy adults to examine the relationships of chronotype and depression with the proportions of REM and deep sleep [20]. We used a similar approach and also used a combination of questionnaire survey and Fitbit sleep phase analysis in a relatively small cohort of young healthy adults. However, in our study, we differentiated between workdays and days off. Our aim was to elucidate the relationships between REM and deep sleep, objective measures of sleep timing and subjective sleepiness, to better understand the background of the discrepancy between sleep quality on workdays and days off.

## 2. Materials and Methods

### 2.1. Ethics, Study Design and Sample Characteristics

The questionnaire surveys and the assessments of Fitbit sleep data in healthy subjects were performed in accordance with the ethical requirements of the Declaration of Helsinki and were approved by the Research Ethics Committee of the Medical Faculty (ChronoSleep study consent number: 2019-3786). All subjects provided informed consent after receiving a complete description of the study. Exclusion criteria were (1) age below 18 years, (2) shift work, (3) work on weekends, (4) chronic diseases including sleep disorders treated with sleep medication and (5) chronic medication including sleep medication. Part of the questionnaire data used for this study came from the same dataset of medical students in their first year, as described earlier [21]. For this dataset, medical students in their first year were invited to participate between 25 June 2021 and 19 May 2022. During this period, due to the COVID-19 pandemic, most of the lectures that were made available as screencasts were watched by students on their own schedule, but usually in the morning before the on-site courses began at 11 a.m. Also, most bars and restaurants were closed during this period due to COVID-19 shutdown, limiting the opportunity for student gatherings.

Volunteers received a pseudonym and were equipped with a Fitbit Inspire multisensory (motion and heart rate) sleep tracking device and asked to wear it for 90 days; hence 64 weekdays = workdays and 26 weekend days = free days, as continuously as possible, especially at nights. After the data collection period, these volunteers were asked to actively participate in the study by filling out the online questionnaire and authorizing sleep data transfer from Fitbit to our study server. All questions in the survey were mandatory and could only be answered once. Data from the questionnaire and Fitbit were assigned by pseudonym. Since we did not collect the names of the study participants, data collection was carried out anonymously. 

From the 90 participants fulfilling the criteria, 31 were excluded because they did not authorize Fitbit data transfer or because they had missing Fitbit data for more than 35 days. This resulted in a final sample size of *n* = 59. In this sample, an average of 82.4 (±9.7) days of Fitbit sleep data were recorded. This corresponds to an average of 988.8 (±116.4) night hours (between 8 p.m. and 8 a.m.).

The online questionnaire included items on age, biological sex, as well as on tiredness in the morning (TN) on workdays and free days, sleep disturbance (Dis) and the use of an alarm clock on workdays and free days during the last 3 month (=data collection period). The average age was 21.18 (±2.36) years.

According to the questionnaire, all subjects used an alarm clock on workdays, while only fifteen subjects used one on free days. This shows that for most subjects, the end of sleep on workdays was determined by the alarm clock, even though they were relatively flexible about when the started in the morning.

Table 1 provides a definition of the scores and the distribution of the variables assessed by questionnaire.

### 2.2. Assessment of Objective Measures of Sleep Timing, Sleep Architecture and Circadian Rhythms

Objective measures for sleep timing, variables of sleep architecture and analyses of circadian rhythms in sleep stages and were calculated based on Fitbit sleep data. Schematic representation of sleep stage analyses, as displayed by the proprietary Fitbit application, illustrates sleep timing and sleep stage architecture (Figure 1). A custom software application was developed using Python 3.10 to process Fitbit sleep data accordingly and separately for workdays and free days.

Objective measures for sleep timing include midpoint of sleep in clock time (MS) and sleep duration (SD). Social jet lag (SJL) was calculated by the difference between MS on free days and workdays [deltaMS = MS free days − MS workdays]. Sleep loss (SL) was calculated by the difference between sleep duration on free days and workdays [deltaSD = SD free days − SD workdays] [22]. 

Circadian rhythm analyses of Fitbit sleep data were performed using ClockLab 6.1.10 toolbox (Actimetrics, Wilmette, IL, USA). For the graphical representation of longitudinal sleep stage data, actograms were created at a 60 s resolution. Acrophases of sleep stage rhythms were calculated by fitting the respective data of each day to a sine function with a period of 24 h and its average phase was calculated using the phase angle method. Chi squared periodogram analysis was performed using a confidence level of 0.001, where the amplitude reflects the consistency of the waveform at the respective period, 12 h or 24 h. Thus, amplitude is a measure for circadian rhythm stability. Circadian rhythmicity was confirmed using Morlet continuous wavelet transform analyses [23].

Variables of sleep architecture include amount (min) of total sleep, light sleep, deep sleep, REM sleep and wake after sleep onset and proportion of light sleep, deep sleep, REM sleep and wake after sleep onset (% of total sleep). In addition, fragmentation index of REM sleep (RFI), deep sleep (DFI) and wake after sleep onset (WFI) were calculated by the number of transitions from the respective stage to any other stage per hour of the respective stage [16], e.g., RFI = number of transitions from REM sleep to any other sleep stage per hour of REM sleep. 

### 2.3. Statistical Analyses 

Statistical analyses were performed using Graph Pad Prism 7.01 software. *p* values < 0.05 were considered statistically significant. Normality was tested by D’Agostino-Pearson normality test. Differences between workdays and free days were analysed by paired *t*-tests if both variables were normally distributed, otherwise by Wilcoxon test. To investigate the direction of relationships among the variables, bivariate correlations were computed for Spearman’s rank coefficients with a 95% confidence interval.

## 3. Results

### 3.1. Subjective Measures of Sleep Quality and Objective Measures of Sleep Timing

Importantly, tiredness on waking, as a measure for subjective sleep quality, was significantly higher on workdays than on free days (data not normally distributed; Wilcoxon test, *p* < 0.0001). The frequency distribution illustrates the shift to lower subjective tiredness on free days (Figure 2a). This shift probably reflects the importance of the freedom to sleep in as long as desired for subjective sleep quality and is in agreement with a larger cohort study showing a higher subjective sleep quality on free days [9].

On free days, the midpoint of sleep (MS) was significantly later (Figure 2b), and sleep duration (SD) was significantly longer (Figure 2c) than on workdays. This is consistent with a shift in sleep timing and duration between biological timing conditions (free days) and social timing conditions (e.g., alarm clock on workdays), resulting in social jet lag (SJL) and sleep loss (SL), respectively (Figure 2c). 

Correlations between the variables assessed by questionnaire and objective measures for sleep timing are summarized in Figure 3. As expected, later MS on free days, consequently the later chronotype, is associated with a shorter SD on workdays, and consequently with higher SL. This is consistent with the earlier wake-up time induced by the alarm clock on workdays. A higher score in self-reported tiredness on free days (but not on workdays) is associated with a higher score of sleep disturbance. This suggests that tiredness on free days is a better indicator of general impairment of sleep quality than tiredness on workdays. Surprisingly, self-reported sleep disturbance is associated with longer sleep duration on workdays, suggesting a compensatory higher need for more sleep.

### 3.2. Circadian Rhythms of Sleep Stages

Representative longitudinal recordings of total sleep and sleep stages are shown in Appendix A, respectively. A representative analysis of the circadian rhythms is shown in Appendix A. Wavelet transformation confirms regular oscillations of sleep stages around 24 h (Appendix A), indicating circadian rhythmicity. Consistently, periodograms reveal circadian oscillations of sleep stages (Appendix A). In addition, ultradian (around 12 h) oscillations are observed regularly (Appendix A), albeit with smaller amplitude than the circadian oscillations. 

The acrophase of the circadian rhythm in deep sleep is 1.6 (±0.3) hours earlier, and the acrophase of the circadian rhythm in REM sleep is 0.86 (±0.4) hours later than the acrophase of the circadian rhythms of total sleep (Figure 4a). This indicates a higher prevalence of deep sleep during early sleep and of REM sleep during late sleep. Consistently, there is a close phase relationship between the sleep stages and total sleep in most subjects (Figure 4b). Interestingly, the amplitudes of the circadian (24 h) rhythm are higher for light sleep and deep sleep than of REM sleep and wake after sleep onset (Figure 4c). This indicates that the rhythm of deep sleep is more stable than the rhythm of REM sleep.

Correlations of variables assessed by questionnaire and the measures of objective sleep timing with the phase and the amplitude of the circadian rhythm of sleep stages are summarized in Figure 5. As expected, the phase of the circadian rhythms of all sleep stages, including wake after sleep onset, correlates positively with MS. Surprisingly, male sex is associated with an earlier phase and a lower amplitude in deep sleep. Interestingly, longer SD on workdays is associated with an earlier phase in the circadian rhythms of deep sleep and wake after sleep onset. More importantly, shorter sleep duration on workdays and consistently higher sleep loss are associated with lower amplitude in the circadian rhythm of REM sleep. This indicates that chronotype-dependent sleep loss affects stability of the REM sleep rhythm. 

### 3.3. Sleep Architecture

The difference in sleep composition on workdays and free days is shown in Figure 6. The amount of total sleep, light sleep, and deep sleep is higher on free days than on workdays (Figure 6a). Thus, a longer SD on days off is mostly composed of light and deep sleep. However, the proportion of sleep stages and wake after sleep onset relative to total sleep did not differ between workdays and free days (Figure 6b). Furthermore, the fragmentation of deep sleep (Figure 6c), REM sleep (Figure 6d) and wake after sleep onset (Figure 6e) did not differ between workdays and free days. 

The correlations between the sleep composition parameters are summarized in Figure 7a. The amount of one sleep stage per night correlates positively with the respective proportion. In addition, the proportion of deep sleep correlates negatively with the proportion of light sleep, which suggests an inverse relationship between deep sleep and light sleep. The proportions of deep sleep and REM sleep correlate negatively with the fragmentation of the respective sleep stage. In contrast, the proportion of light sleep correlates positively with the fragmentation of deep sleep and REM sleep. In addition, the proportion of wake after sleep onset correlates positively with the fragmentation of REM sleep. Thus, higher proportions of deep sleep and REM sleep are associated with a better consolidation of the respective sleep stages, while higher proportions of light sleep are associated with a higher fragmentation of deep sleep and REM sleep. Moreover, higher proportions of wake after sleep onset are associated with a higher fragmentation of REM sleep. 

The correlations of sleep composition parameters with the amplitude of circadian rhythm of sleep stages are summarized in Figure 7b. The amplitudes of deep sleep and REM sleep correlate positively with the proportion of the respective sleep stage and negatively with the proportion of light sleep and RFI. In addition, the amplitude of REM sleep correlates positively with the proportion of deep sleep. Thus, a higher proportion of deep sleep, a lower proportion of light sleep and a better consolidation of REM sleep are associated with a higher robustness of the circadian rhythms of deep sleep and REM sleep.

The correlations between variables assessed by questionnaire, objective measures for sleep timing and the most relevant sleep composition variables are summarized in Figure 8. On free days, the proportion of deep sleep on free days correlates negatively with SD and SL. In addition, SL is associated with a higher proportion of light sleep and, consistently, with a higher fragmentation of REM sleep. Thus, the need for higher SD on days off and/or a higher discrepancy between SD on workdays and days off is most likely predicted by an intrinsically determined lower proportion of deep sleep, a higher proportion of light sleep and a higher fragmentation of REM sleep. However, SD on workdays correlates positively with the proportion of REM sleep on free days. This suggests that the intrinsically determined individual proportion of REM may predict the need for longer SD on workdays. Moreover, male biological sex correlates with a lower proportion of REM sleep on free days. Importantly, subjective tiredness upon waking on free days and self-reported sleep disturbances correlate negatively with the proportion of REM sleep and consistently correlate positively with the proportion of light sleep and RFI on workdays. This suggests that the proportion of light sleep and REM sleep, as well as RFI, are promising digital markers for general poor sleep quality.

## 4. Discussion

This study of longitudinal data on sleep stages in young healthy adults under real-life condition provides important insight into the relationship between circadian rhythms and the composition of sleep under social timing conditions/workdays and biological timing conditions/days off. 

In our cohort, the chronotype showed a normal variation consistent with previous data for young healthy adults [8,24]. As expected, the midpoint of sleep was significantly earlier and sleep duration was significantly shorter on workdays than on free days. Consequently, later chronotype was associated with both higher social jet lag and sleep loss. Students may nap to compensate for sleep loss [25] and daytime naps affect nocturnal sleep efficiency [26]. Future studies are needed to explore how naps on workdays and free days affect nocturnal sleep architecture. 

Subjective tiredness upon waking was higher on workdays than on free days, which aligns with the results from a larger cohort study showing higher sleep quality on days off [11]. Feeling rested upon waking is not only important for defining subjective sleep quality [27] but also for well-being and performance. Therefore, this study, like our previous one [21], supports that considering endogenous sleep time in working life can have a positive impact on performance and general health. In contrast, tiredness on free days, when the sleep timing is mostly determined by biological/intrinsic factors, correlated positively with self-reported sleep disturbance. This suggests that tiredness on free days is a better indicator of sleep quality determined by intrinsic factors than tiredness on workdays. This highlights the importance of modifying the standardized sleep questionnaires by differentiating between workdays and free days to improve sensitivity and specificity, as suggested previously [11]. 

Circadian rhythm analyses show regular oscillation of sleep stages and wake after sleep onset around 24 h (circadian) and additional oscillations around 12 h (ultradian), albeit with lower amplitude, indicating lower robustness. This is consistent with earlier studies on circadian and ultradian oscillations in sleep [28]. Moreover, the data from this study are consistent with longitudinal analysis of core body temperature [29], indicating that under real-life conditions, sleep and other circadian body rhythms are not tightly entrained to 24 h but oscillate in a circadian manner. The phase of the circadian rhythms in the sleep stages correlate positively with the midpoint of sleep. Thus, sleep timing affects the timing of the sleep stages. The phase of circadian rhythm in deep sleep was about three hours earlier than the phase of REM sleep, consistent with the higher prevalence of deep sleep and REM sleep during the first and second half of the night, respectively [12,13]. However, little is known about whether the timing of REM and deep sleep is linked to sleep onset or to usual bedtime and/or to each other. In other words, is the acrophase of REM sleep a function of the elapsed sleep time, the time of day, or the acrophase of deep sleep?

Correlations of the phases and amplitudes of circadian rhythms in deep sleep and REM sleep with the proportions and fragmentations of the respective sleep stages suggest particular relationships. Based on the higher circadian rhythm amplitude and lower fragmentation, deep sleep appears, overall, to be more robust than REM sleep. However, the proportion of deep sleep and the amplitude of REM sleep correlated positively, suggesting that deep sleep supports REM sleep rhythm stability. The amount of light sleep and deep sleep was higher on free days than on workdays, indicating that the longer sleep duration on free days is mostly composed of these sleep stages. Interestingly, male sex was associated with an earlier phase and a lower amplitude of the circadian rhythm of deep sleep, and with a lower proportion of REM sleep on free days. The data correspond with earlier findings regarding differences in sleep architecture [20,30,31] and circadian rhythms [32] between sexes. However, females are still underrepresented in studies on circadian rhythms and sleep [32]. A recent review discussed the data gap on the effects of sex on human circadian physiology and sleep physiology, as well as the needs for future research to include sex as an important factor [33].

Importantly, shorter sleep duration on workdays was associated with a later phase of circadian rhythm in deep sleep, lower amplitude of circadian rhythm in REM sleep and shorter proportion of REM sleep on free days. Similarly, sleep loss was associated with a lower amplitude, a higher fragmentation of REM sleep and a lower proportion of deep sleep on free days. In contrast, higher self-reported tiredness on free days and sleep disturbance were associated with a lower proportion of REM sleep and higher fragmentation of REM sleep. Thus, in particular RFI, a parameter that has so far been assessed only by polysomnography in the context of REM sleep behavior disorders [16], is a promising digital marker for the objective assessment of general poor sleep quality/sleep disturbances. Our observations are consistent with REM sleep appearing to be the sleep stage most relevant to sleep quality and cognitive performance [34] but also most sensitive to disruption [35]. However, further studies on longitudinal sleep stage recordings in larger cohorts are needed to establish benchmarks for sleep architecture.

In summary, our study suggests that REM sleep proportion and consolidation not only play a key role in sleep quality but are also affected by the misalignment between external (workdays) and internal (free days) sleep timing (Figure 9). Our results suggest that longitudinal recording of sleep stages, particularly REM sleep, represents an exciting area for research into digital markers of sleep quality.

## 5. Limitations of the Study

As polysomnography does not allow for longitudinal studies under real-life conditions, the sleep stages were analysed by the proprietary Fitbit algorithms, which have a lower accuracy. However, we would like to emphasize that this study is not intended to promote Fitbit sleep analysis as a diagnostic tool in sleep medicine.

The sample of this study is rather small, not representative of the population, and has biases in many ways, such as age, education, income, biological sex and motivation to participate in our study. 

The study is very special in terms of the study period, which fell within the COVID-19 pandemic. On the one hand, the subjects were relatively flexible about when they started in the morning. However, all subjects used an alarm clock on weekdays, which is probably due to the fact that the workload for first-year medical students is very high. Nevertheless, differences between weekdays and days off are likely to be much larger when working hours are rigid. On the other hand, confounding factors for midsleep on free days, such as social gatherings, were largely reduced due to the general closure of most of the restaurants, bars and nightclubs. 

Correlations were used as a measure of the direction of relationships, but do not imply causation. 

## Figures and Tables

**Figure 1 brainsci-13-01482-f001:**
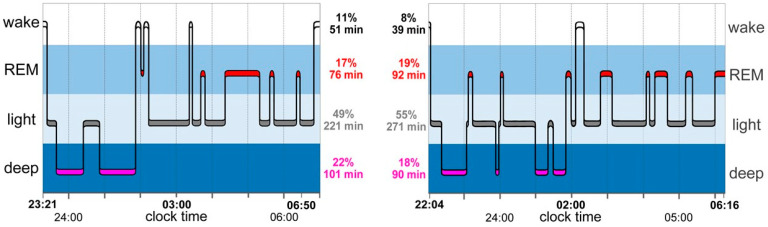
Schematic representation of sleep data for a subject on two consecutive nights, as displayed by the proprietary Fitbit application. We used a custom application to extract objective measures of sleep timing such as midpoint of sleep and sleep duration and variables of sleep architecture such as amount, percentage and fragmentation of sleep stages. The Clocklab toolbox (Actimetrics) was used for the analyses of phase and amplitude of the circadian rhythm of the sleep stages.

**Figure 2 brainsci-13-01482-f002:**
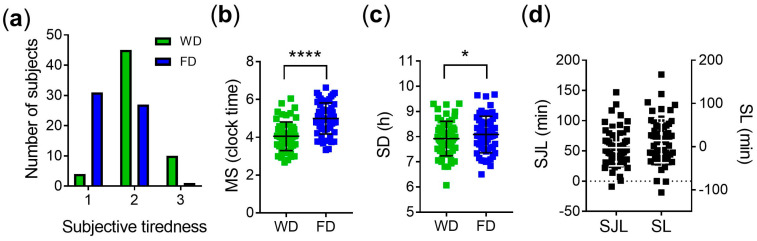
Differences in subjective sleep quality and objective sleep timing between workdays (WD) and free days (FD). (**a**) Frequency distribution of self-reported tiredness upon waking (1, well rested, 2 tired, 3 very tired), as a measure for subjective sleep quality. (**b**) Midpoint of sleep (MS). Data normally distributed, *t*-test **** *p* < 0.0001. (**c**) Sleep duration (SD). Data normally distributed, *t*-test * *p* < 0.0001; (**d**) Differences in MS and SD between WD and FD result in social jet lag (SJL) and sleep loss (SL), respectively. Middle lines and error bars represent the means ± standard deviation.

**Figure 3 brainsci-13-01482-f003:**
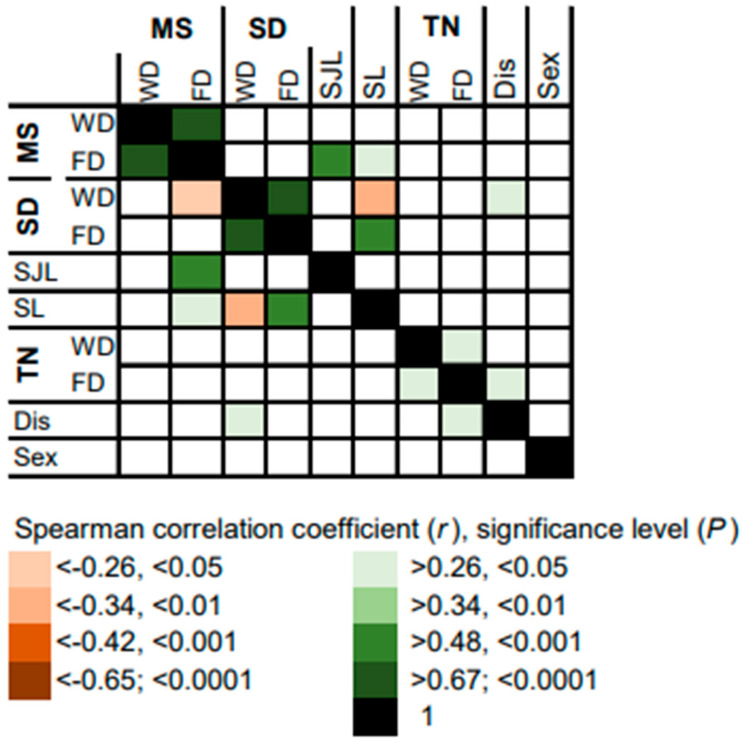
Heatmap summarizing correlations between measures for objective sleep timing and variables assessed by questionnaire. Midpoint of sleep (MS), sleep duration (SD), social jet lag (SJL), sleep loss (SL), self-reported tiredness upon waking (TN), self-reported sleep disturbance (Dis) and biological sex. Where applicable, the data are expressed separately for workdays (WD) and free days (FD).

**Figure 4 brainsci-13-01482-f004:**
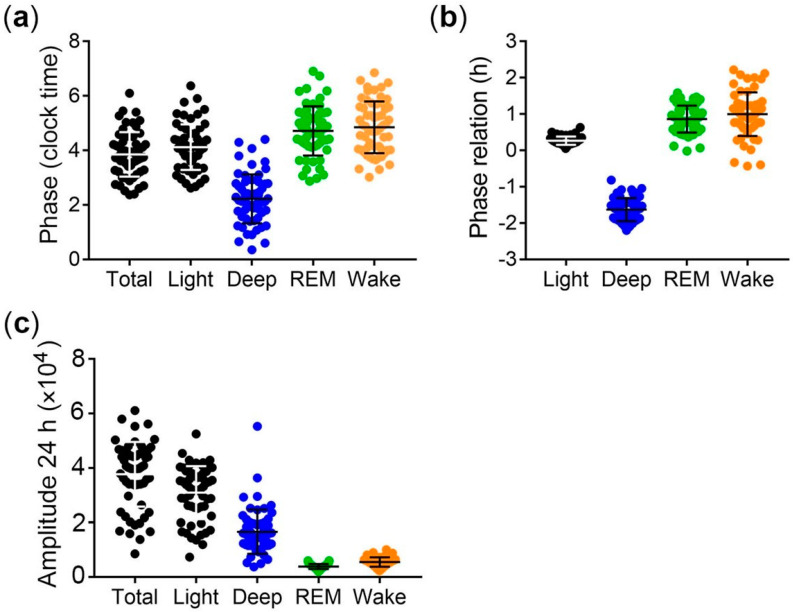
Phase and amplitude of circadian rhythms of sleep stages. (**a**) Phase of the circadian rhythms of total sleep and sleep stages; (**b**) phase of the circadian rhythm in sleep stages relative to the phase of the circadian rhythm in total sleep; (**c**) amplitudes of the circadian (24 h) rhythms of total sleep and sleep stages. Middle lines and error bars represent the means ± standard deviation.

**Figure 5 brainsci-13-01482-f005:**
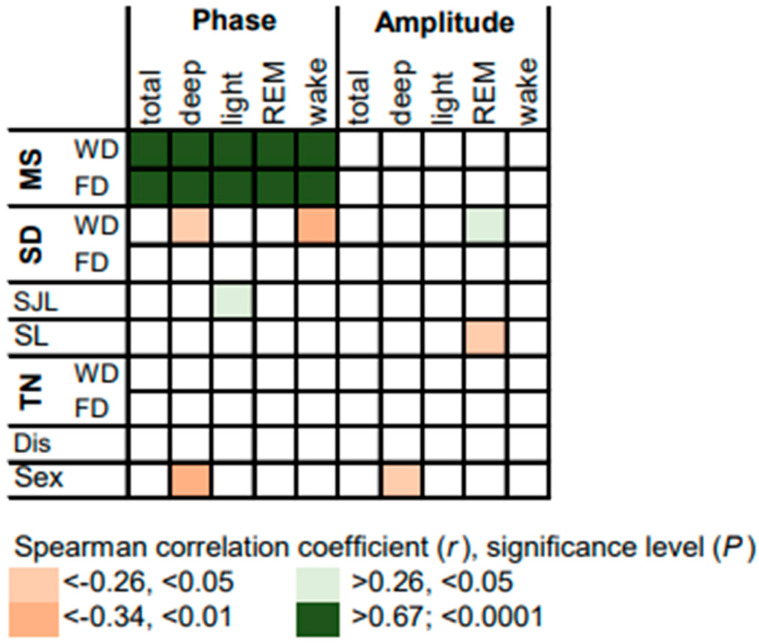
Heatmap summarizing correlations between variables assessed by questionnaire, objective measures of sleep timing and variables on circadian rhythms of sleep stages. Midpoint of sleep (MS), sleep duration (SD), social jet lag (SJL), sleep loss (SL), self-reported tiredness upon waking (TN), self-reported sleep disturbance (Dis), biological sex (Sex). Where applicable, the data are expressed separately for workdays (WD) and free days (FD).

**Figure 6 brainsci-13-01482-f006:**
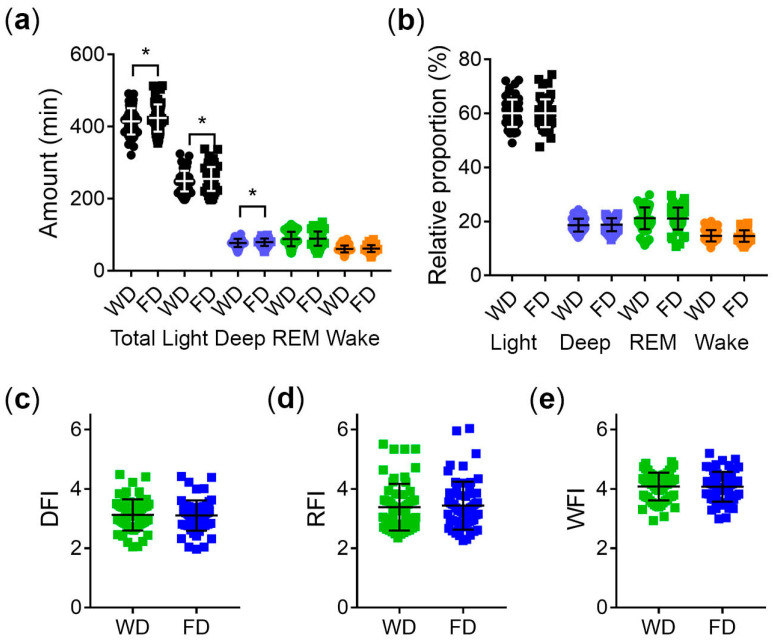
Differences in sleep composition between workdays (WD) and free days (FD): (**a**) amount of sleep stages per night in minutes. Data normally distributed, *t*-test * *p* < 0.05; (**b**) proportion of sleep stages in percent of total sleep; (**c**) deep sleep fragmentation index (DFI); (**d**) REM sleep fragmentation index (RFI); and (**e**) fragmentation index of wake after sleep onset (WFI). Middle lines and error bars represent the means ± standard deviation.

**Figure 7 brainsci-13-01482-f007:**
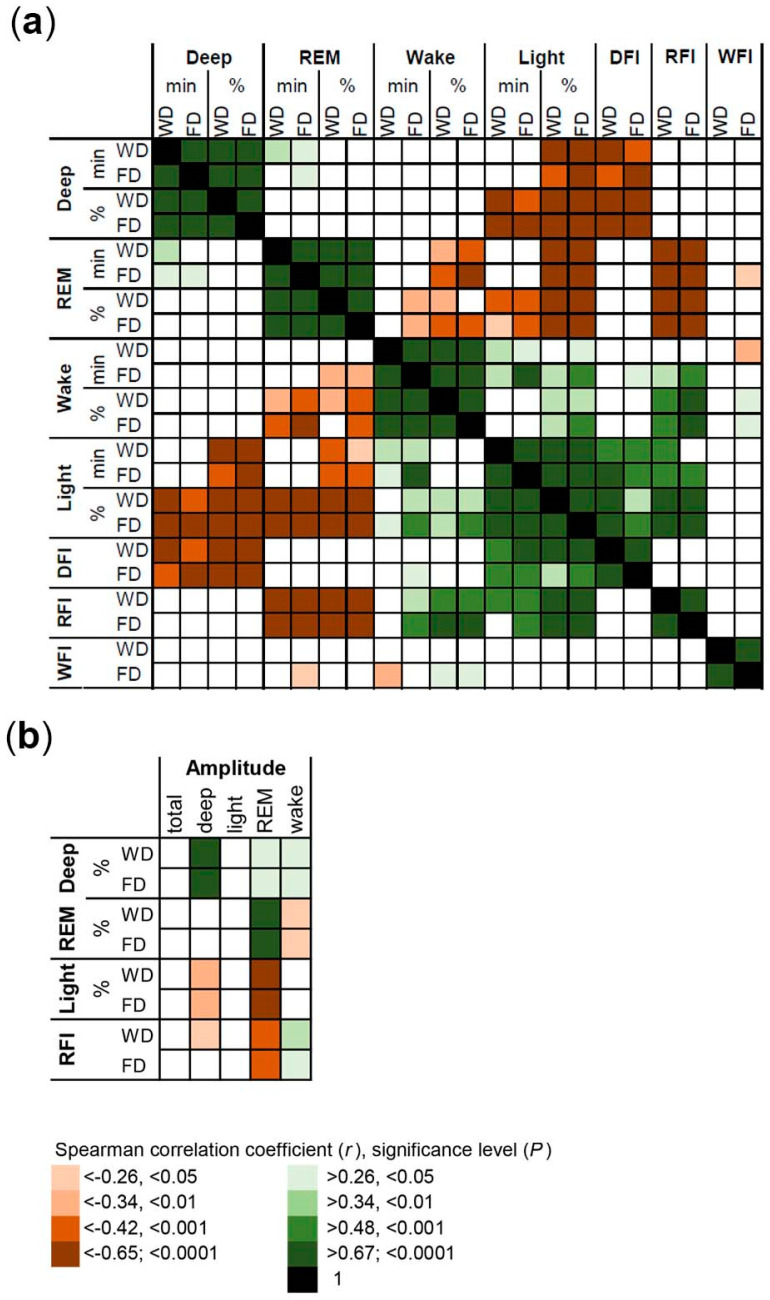
Heatmaps summarizing the correlations between variables of sleep composition and the amplitude of the circadian rhythms of sleep stages. (**a**) Correlations between sleep stage amount (min) and proportion (% of total sleep), fragmentation index of deep sleep (DFI), fragmentation index of REM sleep (RFI) and fragmentation index of wake after sleep onset (WFI). (**b**) Correlations between amplitude of the circadian rhythms in sleep stages with the most relevant variables of sleep. Data are expressed separately for workdays (WD) and free days (FD).

**Figure 8 brainsci-13-01482-f008:**
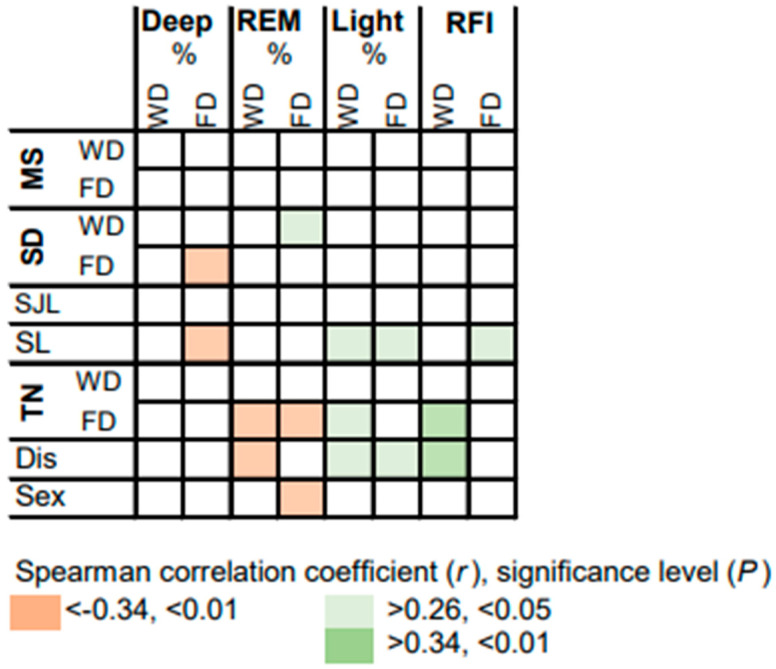
Heatmap summarizing correlations between measures of objective sleep timing, self-reported tiredness on waking, sleep disturbances, biological sex and most relevant variables on sleep architecture. Midpoint of sleep (MS); sleep duration (SD); social jet lag (SJL); sleep loss (SL); self-reported tiredness on waking (TN); self-reported sleep disturbance (Dis); biological sex (Sex); proportions of deep sleep, REM sleep and light sleep (in % of total sleep); REM sleep fragmentation index (RFI). Where applicable, the data are expressed separately for workdays (WD) and free days (FD).

**Figure 9 brainsci-13-01482-f009:**
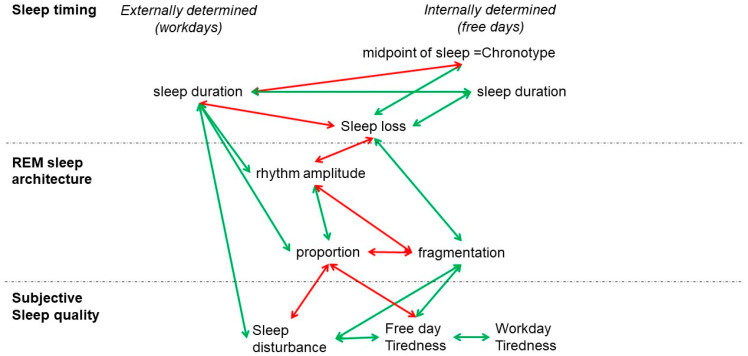
Graphic summary. Chronotype-related sleep loss is correlated with the amplitude of the circadian rhythm and the fragmentation of REM sleep. The amplitude of the circadian rhythm and the fragmentation of REM sleep are correlated with the proportion of REM sleep. Self-reported tiredness on free days and sleep disturbances are correlated with the proportion and fragmentation of REM sleep. Green arrows indicate positive correlations and red arrows indicate negative correlations.

**Table 1 brainsci-13-01482-t001:** Definition and distribution of vvariables assessed by questionnaire.

Variable			Score	Amount (Proportion)
Sex				
	Female		1	43 (73%) ^1^
	Male		2	16 (27%)
Tiredness on waking				
	Workdays			
		Well rested	1	4 (7%)
		Tired	2	45 (76%)
		Very tired	3	10 (17%)
Free days				
		Well rested	1	31 (52%)
		Tired	2	27 (46%)
		Very tired	3	1 (2%)
Sleep disturbance				
	No		1	55 (93%)
	Yes		2	4 (7%)

^1^ The high proportion of females reflects the distribution of biological sex among all first-year medical students.

## Data Availability

Data are unavailable due to privacy or ethical restrictions.

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
