# Peer review of "Chronotype-Dependent Sleep Loss Is Associated with a Lower Amplitude in Circadian Rhythm and a Higher Fragmentation of REM Sleep in Young Healthy Adults"

_brainsci, 2023, doi:10.3390/brainsci13101482_

Round 1
Reviewer 1 Report
This is an interesting, well-designed, and practical study. I have no particular concern with this research paper. Only two minor points for Discussion:
1- The authors said: “Moreover, the data of this study are consistent with longitudinal analysis of core body temperature [27], indicating that under real-life conditions, sleep and other circadian body rhythms are not tightly entrained to 24 hours but oscillate in a circadian manner. The phase of the circadian rhythms in the sleep stages correlate positively with the midpoint of sleep. Thus, sleep timing affects the timing of the sleep stages.”. Can the authors explain more? What does it really mean: “sleep timing affects the timing of the sleep stages”.
2- The authors said: “Interestingly, male sex was associated with an earlier phase and a lower amplitude of the circadian rhythm of deep sleep, and with a lower proportion of REM sleep on free days. The data correspond with earlier findings regarding differences in sleep architecture [20,28,29] and circadian rhythms [30] between sexes. However, females are still underrepresented in studies on circadian rhythms and sleep [30]”. Can the authors provide more hypotheses for this sex difference?
Author Response
This is an interesting, well-designed, and practical study. I have no particular concern with this research paper.
Response: Thank you very much.
Only two minor points for Discussion:
1- The authors said: “Moreover, the data of this study are consistent with longitudinal analysis of core body temperature [27], indicating that under real-life conditions, sleep and other circadian body rhythms are not tightly entrained to 24 hours but oscillate in a circadian manner. The phase of the circadian rhythms in the sleep stages correlate positively with the midpoint of sleep. Thus, sleep timing affects the timing of the sleep stages.”. Can the authors explain more? What does it really mean: “sleep timing affects the timing of the sleep stages”.
Response: We explained this in more detail and added the following: “However, little is known about whether the timing of REM and deep sleep is linked to sleep onset or to usual bedtime and/or to each other. In other words, is the acrophase of REM sleep a function of the elapsed sleep time, the time of day, or the acrophase of deep sleep?”
2- The authors said: “Interestingly, male sex was associated with an earlier phase and a lower amplitude of the circadian rhythm of deep sleep, and with a lower proportion of REM sleep on free days. The data correspond with earlier findings regarding differences in sleep architecture [20,28,29] and circadian rhythms [30] between sexes. However, females are still underrepresented in studies on circadian rhythms and sleep [30]”. Can the authors provide more hypotheses for this sex difference?
Reviewer 2: We added the following: “A recent review discussed the data gap on the effects of sex on human circadian physiology and sleep physiology, as well as the needs for future research to include sex as an important factor [31].”
Reviewer 2 Report
The manuscript entitled " A higher discrepancy between sleep duration on workdays and free days is associated with a reduced amplitude in circadian rhythm and a higher fragmentation of REM sleep in young healthy adults" addresses a highly significant and timely topic. Real-life studies of sleep/wake rhythms in home-based conditions are very useful for monitor sleep in natural conditions and filling the gaps in knowledge about sleep. Authors used the Fitbit device, which allows authors to describe sleep stages, as well.
The article is very hard to follow. I think the article was written very quickly. There are more typos (starting in the title with the word discrepancy).
Some important information you must read in another article, what I find somehow rude and not scientifically correct – for example:
L73-76 - if you want to understand the aims of the study, please read article 20;
L80-82 – read article 21 to see, how many participants etc. we had.
The Title – I think the Title of the article is too long and very specific to one result. I will recommend authors to make the title shorter.
What do authors mean by „reduced amplitude in circadian rhythm“, what result is it in what figure?
The same sentence as the title of the article is in the abstract L20-22.
Abstract – there is no evidence about the study group and study duration.
Introduction:
Page 2, L 73: the aim of the study needs to be rewritten.
The last sentence in the introduction is perhaps better to have at the end of the discussion.
Material and Methods:
Needs to be rewritten with the information about the study participants, inclusion and exclusion criteria and study design. The authors write description information in the first paragraph of the results, but I think the information belongs to the materials section, including Table 1.
Table 1 is hard to read, what is the meaning of § in the table?
Results:
Figure 2: Why did you choose clock time for MS and h for SD?
How did you determine the chronotype of participants? Did you use some questionnaires or only according to the midpoint of sleep on free days? Do you think it is sufficient when some participants have quite high SJL on FD?
What are the middle lines (average, mesor) and the deviations (SD, SEM) in graphs? Please, write it into the figure legend. Graphs Fig.6 c,d,e have no deviations and some formal mistakes are in figure legend.
L-179 – part 3.2
Reorganize the paragraphs, I think the main results should be the first, then supplementary materials.
Figure 4 – I think you should change the colour of black - white graph on the white background, it is not good visible.
Fig. S1: - make the figure legend more readable. And the same question as before - How did you find out the chronotypes of the selected subjects?
Fig S3: - The figure legend is very poor; it is not common knowledge to understand periodograms and amplitudes.
In general, the figure legends in the supplement are different from the previous ones, please use semicolons between figure descriptions.
The main idea I was thinking of from the beginning and the authors mention in the discussion L322-327 very correctly, there are sex differences in sleep parameters. Also, you have in results some correlations to worsen sleep in men. I think it should be very useful to look separately at women and men. There is a lack of studies performed on women as you also mention in your discussion “ However, females are still underrepresented in studies on circadian rhythms and sleep“ L 326.
Author Response
The manuscript entitled " A higher discrepancy between sleep duration on workdays and free days is associated with a reduced amplitude in circadian rhythm and a higher fragmentation of REM sleep in young healthy adults" addresses a highly significant and timely topic. Real-life studies of sleep/wake rhythms in home-based conditions are very useful for monitor sleep in natural conditions and filling the gaps in knowledge about sleep. Authors used the Fitbit device, which allows authors to describe sleep stages, as well.
Response: Thank you very much.
The article is very hard to follow. I think the article was written very quickly. There are more typos (starting in the title with the word discrepancy).
Response: We apologize for typos. The article will undergo additional language editing.
Some important information you must read in another article, what I find somehow rude and not scientifically correct – for example:
L73-76 - if you want to understand the aims of the study, please read article 20;
Response: We referred to article 20 as we used a similar approach namely the combination of questionnaire and Fitbit sleep stage analyses. We have clearly described the aim of the study in this paragraph: “…to explore relationships between rhythms, proportions and fragmentations of REM- and deep sleep with objective measures of sleep timing and subjective tiredness separately for workdays and days off”
L80-82 – read article 21 to see, how many participants etc. we had.
Response: We referred to article 21 as we used the same dataset. The sample characteristics are described in the result section, L143-151: “As mentioned above, the general sample characteristis are described elsewhere [21]. Briefly, from the 90 participants fullfilling the inclusion and exclusion criteria 17 were excluded because they did not authorize Fitbit data transfer or because of missing Fitbit data for more than 35 days. This resulted in a final sample size of 59. In this sample, an average of 82.4 (+/-9.7) days of Fitbit sleep data were recorded. The average age was 21.18 (+/-2.36) years. All subjects used an alarm clock on workdays, while only 17 subjects used one on free days. The variables assessed by questionnaire which are relevant for this study are shown in table 1. A small proportion (4%) of subjects reported sleep disturbances.”
The Title – I think the Title of the article is too long and very specific to one result. I will recommend authors to make the title shorter.
What do authors mean by „reduced amplitude in circadian rhythm“, what result is it in what figure?
Response: we now have shortened and rephrased the title: “Chronotype-dependent sleep loss is associated with a lower amplitude in circadian rhythm and a higher fragmentation of REM sleep in young healthy adults.”
The negative correlation between sleep loss (SL) and amplitude in circadian rhythm is shown in figure 5. This indicates that a higher SL is associated with a lower amplitude.
The same sentence as the title of the article is in the abstract L20-22.
Response: right, this is one of the major findings.
Abstract – there is no evidence about the study group and study duration.
Response: We now have included this information to the abstract.
Introduction:
Page 2, L 73: the aim of the study needs to be rewritten. The last sentence in the introduction is perhaps better to have at the end of the discussion.
Response: The aim of the study has been rephrased as follows: “We used a similar approach and also used a combination of questionnaire survey and Fitbit sleep stage analysis in a relatively small cohort of young healthy adults. However, in our study we differentiated between workdays and days off. Our aim was to elucidate the relationships between REM- and deep-sleep, objective measures of sleep timing, and subjective sleepiness, to better understand the background of the discrepancy between sleep quality on workdays and days off.”
Material and Methods:
Needs to be rewritten with the information about the study participants, inclusion and exclusion criteria and study design. The authors write description information in the first paragraph of the results, but I think the information belongs to the materials section, including Table 1.
Table 1 is hard to read, what is the meaning of § in the table?
Response: The table has now been simplified and § has been removed.
Results:
Figure 2: Why did you choose clock time for MS and h for SD?
Response: The midpoint of sleep (MS) is by definition a clock time while the sleep duration (SD) is by definition a period of time.
How did you determine the chronotype of participants? Did you use some questionnaires or only according to the midpoint of sleep on free days? Do you think it is sufficient when some participants have quite high SJL on FD?
Response: The midpoint of sleep on free days defines the chronotype (see introduction, references 4,5). As described in the Materials and Methods section, the MS is one of the objective measures for sleep timing, which has been calculated from Fitbit sleep data. Therefore, the MS is not only objective but also more accurate than if collected through questionnaires because it is based on repeated measurements. We added the information on MS in the legend of figure 1.
What are the middle lines (average, mesor) and the deviations (SD, SEM) in graphs? Please, write it into the figure legend.
Response: The definition of middle lines and bars have now been added to the legends.
Graphs Fig.6 c,d,e have no deviations and some formal mistakes are in figure legend.
Response: Error bars have now been included and formatting errors corrected.
L-179 – part 3.2
Reorganize the paragraphs, I think the main results should be the first, then supplementary materials.
Response: The supplement material serves to illustrate the data used for the analyses and is therefore a basis for the main results. Therefore it appears first.
Figure 4 – I think you should change the colour of black - white graph on the white background, it is not good visible.
Response: The color code is used to illustrate the different sleep stages. On our screen/printer all colors are very easy to see, especially since the mean and SEM are also highlighted.
Fig. S1: - make the figure legend more readable. And the same question as before - How did you find out the chronotypes of the selected subjects?
Response: We changed the figure legend and included the definition of chronotype
Fig S3: - The figure legend is very poor; it is not common knowledge to understand periodograms and amplitudes.
Response: The figure legend has been revised and expanded to explain the circadian rhythm analyses such as periodograms.
In general, the figure legends in the supplement are different from the previous ones, please use semicolons between figure descriptions.
Response: We now have used semicolons consistently.
The main idea I was thinking of from the beginning and the authors mention in the discussion L322-327 very correctly, there are sex differences in sleep parameters. Also, you have in results some correlations to worsen sleep in men. I think it should be very useful to look separately at women and men. There is a lack of studies performed on women as you also mention in your discussion “ However, females are still underrepresented in studies on circadian rhythms and sleep“ L 326.
Response: We totally agree that it is important to look separately at women and men. This is why we included the variable “sex” in our correlation analyses.
Reviewer 3 Report
Comments:
This is a manuscript evaluating sleep duration, circadian rhythm and higher sleep fragmentation in workdays and free days. The research question is interesting but is not novel. This is a follow up part of a previously published paper, where the authors describe the findings of the wearable FitBit. The methods are appropriate. The authors need to address the following queries:
1- The authors have used the data from a previous published paper. They have mentioned that the study was conducted between June 2021 till May 2022 and the participants were First year medical students. The classes were online (screencast) and did not started before 11AM and students had the option to access the class according to their own time schedule. I wonder, in such a situation, what was the definition of “Freedays” (Weekend) and “workdays” (weekdays)? There may be hardly any significant difference between workload these two groups. Moreover, in this manuscript, the authors have not discussed that the study was conducted during this settings and probable implications on the finding of the study. This limits the generalizability of the study and is a major limitation.
2- Therefore, a description of usual expected “Freedays” and “Workdays” activities of the participants should be mentioned.
3- Authors mention that out of 90 participants, 17 were excluded. Then how the final figure of 59 were reached? Other causes of exclusion should be mentioned.
4- In the table 1, Mean has been provided Sex (Male and Female). What does this mean? Sex is a dichotomous variable and mean is not an appropriate measure of expression for such data. Same is for other variable e.g “Sleep disturbances’, ‘Tiredness on waking”.
5- How the tiredness in morning (TN) for days like Saturdays (Which is a weekends), considered? The TN on such days is probably due to poor sleep parameters in Friday night, which in turn may be a result of “Workday” (i.e Friday). The authors should mention how this was considered in the analysis.
6- The statistical analysis and their representation seems to be inappropriate at places. As per my understanding, the authors have categorized the Tiredness on Waking (TN) in to ‘Well rested’, “Tired” and “Very tired”. This is not based on any scale and it is a categorical variable and categorical data are cannot be normally distributed. The differences shown in the Figure 2 (TN Score) are based on Mean and SD which is not the proper mode of representation for categorical variable.
7- Many confounding factors e.g social gatherings, alcohol use, recreational or leisure time activities might have contributed to late mid point of sleep timing during “Freedays”. Authors have not discussed how they have addressed these confounding factors.
8- There is no mention of many sleep parameters, which can provide the basic sleep patterns in the participants. E.g Usual bed time, Waking up time, Day time naps
9- There is no data regarding average hours of use of Fitbit by the participants. Poor adherence with the Fitbit will hamper the internal and external validity of the study.
10- Minor points: Spelling of discrepancy in the Title should be corrected.
11- Minor points: Discrepancy in data about use of alarm clock (Current vs previous), 17 vs 15. Should be corrected.

Minor editing of the english language is required.
Author Response
This is a manuscript evaluating sleep duration, circadian rhythm and higher sleep fragmentation in workdays and free days. The research question is interesting but is not novel. This is a follow up part of a previously published paper, where the authors describe the findings of the wearable FitBit. The methods are appropriate. The authors need to address the following queries:
1- The authors have used the data from a previous published paper. They have mentioned that the study was conducted between June 2021 till May 2022 and the participants were First year medical students. The classes were online (screencast) and did not started before 11AM and students had the option to access the class according to their own time schedule.
2- Therefore, a description of usual expected “Freedays” and “Workdays” activities of the participants should be mentioned.
Response: we have now described the study design again in detail and explained that screencasts were watched by students on their own schedule, but usually in the morning before the on-site courses began at 11 a.m..
And added: “According to the questionnaire, all subjects used an alarm clock on workdays, while only 15 (! Corrected!) subjects used one on free days. This shows that for most subjects, the end of sleep on workdays was determined by the alarm clock, even though they were relatively even though they were relatively flexible about when the started in the morning.
I wonder, in such a situation, what was the definition of “Freedays” (Weekend) and “workdays” (weekdays)?
Response: We now have included the definition of workdays and free days (L97).
There may be hardly any significant difference between workload these two groups. Moreover, in this manuscript, the authors have not discussed that the study was conducted during this settings and probable implications on the finding of the study. This limits the generalizability of the study and is a major limitation.
Response: We addressed the issue “workload” and added the relatively flexible start in the mornings in contrast to rigid work hours to the limitations of the study.
3- Authors mention that out of 90 participants, 17 were excluded. Then how the final figure of 59 were reached? Other causes of exclusion should be mentioned.
Response: We now have included the following: “From the 90 participants fullfilling the criteria 31 were excluded because they did not authorize Fitbit data transfer or because of missing Fitbit data for more than 35 days. This resulted in a final sample size of n=59.”
4- In the table 1, Mean has been provided Sex (Male and Female). What does this mean? Sex is a dichotomous variable and mean is not an appropriate measure of expression for such data. Same is for other variable e.g “Sleep disturbances’, ‘Tiredness on waking”.
Response: Right. We now have removed the means for table 1.
5- How the tiredness in morning (TN) for days like Saturdays (Which is a weekends), considered? The TN on such days is probably due to poor sleep parameters in Friday night, which in turn may be a result of “Workday” (i.e Friday). The authors should mention how this was considered in the analysis.
Response: The tiredness is higher on workdays than on free days! We now have added: “This probably reflects the importance of the freedom to sleep in as long as desired for subjective sleep quality and is in agreement with a larger cohort study showing a higher subjective sleep quality on free days [9].”
6- The statistical analysis and their representation seems to be inappropriate at places. As per my understanding, the authors have categorized the Tiredness on Waking (TN) in to ‘Well rested’, “Tired” and “Very tired”. This is not based on any scale and it is a categorical variable and categorical data are cannot be normally distributed. The differences shown in the Figure 2 (TN Score) are based on Mean and SD which is not the proper mode of representation for categorical variable.
Response: As stated in the legend, we used a non-parametric test (Wilcoxon-test) for comparing the not normally distributed TN on WD and FD. However, we visualized the mean+/-SEM in the graph 1a which was not appropriate. We now have changed the visualization to median with range for the not normally distributed data and specified this in the legend.
7- Many confounding factors e.g social gatherings, alcohol use, recreational or leisure time activities might have contributed to late mid point of sleep timing during “Freedays”. Authors have not discussed how they have addressed these confounding factors.
Response: This is an interesting point which has now been addressed in the paragraph on limitations of the study.
8- There is no mention of many sleep parameters, which can provide the basic sleep patterns in the participants. E.g Usual bed time, Waking up time, Day time naps
Response: The start of sleep and wake-up time are included in the calculation of the MS. Daytime naps is an interesting topic but out of the scope of our study.
9- There is no data regarding average hours of use of Fitbit by the participants. Poor adherence with the Fitbit will hamper the internal and external validity of the study.
Response: We now have included the description of the exclusion of participants based on poor adherence.
10- Minor points: Spelling of discrepancy in the Title should be corrected.
Response: Done. The manuscript will undergo additional language editing.
11- Minor points: Discrepancy in data about use of alarm clock (Current vs previous), 17 vs 15. Should be corrected.
Response: Done. Thank you very much!
Round 2
Reviewer 2 Report
Congratulations on the nice results and article and I wish you good scientific feedback.
Author Response
Thank you very much for the quick, fair and thorough review, which helped to increase the quality of the manuscript.
Reviewer 3 Report
Comments:
The authors have submitted their response to my previous queries. Most of the queries have been addressed adequately. Some minor points still need to be addressed:
1- The information regarding bars, restaurants and social gatherings should not be mentioned in study design. Justification for them should be discussed in discussion or limitations.
2- I do not consider day time naps as out of scope of the research question. Day time naps and it’s duration affect the nocturnal sleep. Authors may refer to the following references:
-Mead MP, Huynh P, Le TQ, Irish LA. Temporal Associations Between Daytime Napping and Nocturnal Sleep: An Exploration of Random Slopes. Ann Behav Med. 2022 Nov 5;56(11):1101-110
-Rea EM, Nicholson LM, Mead MP, Egbert AH, Bohnert AM. Daily relations between nap occurrence, duration, and timing and nocturnal sleep patterns in college students. Sleep Health. 2022 Aug;8(4):356-363.
The authors may discuss about it in their discussion or limitation part.
3- I could not find any data about adherence to Fitbit except for those who are excluded because of non-transfer of data, in the revised manuscript. Information regarding average number of night hours of Fitbit use will be more informative.
4- In Table 1, what do the authors mean by ‘Sleep disturbances”? How it was defined ? especially when the authors mention that chronic diseases including sleep disorders were excluded.
5- Representation of tiredness upon waking (TN) in figure 2 in inappropriate. It may be better expressed as frequency and represented in comparative bar diagram.
6- The limitations needs to be elaborated more rather than justification for using Fitbit. It should discuss about possible flaws, bias in study design.
7- The conclusion should mention about the applicability of the study and future directions.

Minor corrections in the English language needed.
Author Response
The authors have submitted their response to my previous queries. Most of the queries have been addressed adequately. Some minor points still need to be addressed:
1- The information regarding bars, restaurants and social gatherings should not be mentioned in study design. Justification for them should be discussed in discussion or limitations.
Response: we now have addressed this in detail in the limitations.
2- I do not consider day time naps as out of scope of the research question. Day time naps and it’s duration affect the nocturnal sleep. Authors may refer to the following references:
-Mead MP, Huynh P, Le TQ, Irish LA. Temporal Associations Between Daytime Napping and Nocturnal Sleep: An Exploration of Random Slopes. Ann Behav Med. 2022 Nov 5;56(11):1101-110
-Rea EM, Nicholson LM, Mead MP, Egbert AH, Bohnert AM. Daily relations between nap occurrence, duration, and timing and nocturnal sleep patterns in college students. Sleep Health. 2022 Aug;8(4):356-363.
The authors may discuss about it in their discussion or limitation part.
Response: We have now referred to the recommended references on daytime naps in the discussion.
3- I could not find any data about adherence to Fitbit except for those who are excluded because of non-transfer of data, in the revised manuscript. Information regarding average number of night hours of Fitbit use will be more informative.
Response: We have now added the average night hours.
4- In Table 1, what do the authors mean by ‘Sleep disturbances”? How it was defined ? especially when the authors mention that chronic diseases including sleep disorders were excluded.
Response: Sleep disorders are considered a chronic illness if they require medication. We have now added this to the description of the exclusion criteria. In contrast, sleep disturbances refer to a subjective feeling of general poor sleep quality.
5- Representation of tiredness upon waking (TN) in figure 2 in inappropriate. It may be better expressed as frequency and represented in comparative bar diagram.
Response: We have changed figure 2a accordingly.
6- The limitations needs to be elaborated more rather than justification for using Fitbit. It should discuss about possible flaws, bias in study design.
Response: The section on limitations is relatively long and goes far beyond the justification for using Fitbit (L448-458).
7- The conclusion should mention about the applicability of the study and future directions.
Response: The applicability and future directions have been added to the conclusion.
The authors have submitted their response to my previous queries. Most of the queries have been addressed adequately. Some minor points still need to be addressed:
1- The information regarding bars, restaurants and social gatherings should not be mentioned in study design. Justification for them should be discussed in discussion or limitations.
Response: we now have addressed this in detail in the limitations.
2- I do not consider day time naps as out of scope of the research question. Day time naps and it’s duration affect the nocturnal sleep. Authors may refer to the following references:
-Mead MP, Huynh P, Le TQ, Irish LA. Temporal Associations Between Daytime Napping and Nocturnal Sleep: An Exploration of Random Slopes. Ann Behav Med. 2022 Nov 5;56(11):1101-110
-Rea EM, Nicholson LM, Mead MP, Egbert AH, Bohnert AM. Daily relations between nap occurrence, duration, and timing and nocturnal sleep patterns in college students. Sleep Health. 2022 Aug;8(4):356-363.
The authors may discuss about it in their discussion or limitation part.
Response: We have now referred to the recommended references on daytime naps in the discussion.
3- I could not find any data about adherence to Fitbit except for those who are excluded because of non-transfer of data, in the revised manuscript. Information regarding average number of night hours of Fitbit use will be more informative.
Response: We have now added the average night hours.
4- In Table 1, what do the authors mean by ‘Sleep disturbances”? How it was defined ? especially when the authors mention that chronic diseases including sleep disorders were excluded.
Response: Sleep disorders are considered a chronic illness if they require medication. We have now added this to the description of the exclusion criteria. In contrast, sleep disturbances refer to a subjective feeling of general poor sleep quality.
5- Representation of tiredness upon waking (TN) in figure 2 in inappropriate. It may be better expressed as frequency and represented in comparative bar diagram.
Response: We have changed figure 2a accordingly.
6- The limitations needs to be elaborated more rather than justification for using Fitbit. It should discuss about possible flaws, bias in study design.
Response: The section on limitations is relatively long and goes far beyond the justification for using Fitbit (L448-458).
7- The conclusion should mention about the applicability of the study and future directions.
Response: The applicability and future directions have been added to the conclusion.
